# A Framework to Manage Coastal Squeeze

**Rodolfo Silva** [1,*], **María Luisa Martínez** [2], **Brigitta I. van Tussenbroek** [3],
**Laura Odette Guzmán-Rodríguez** [1], **Edgar Mendoza** [1] and **Jorge López-Portillo** [2]

1 Instituto de Ingeniería, Universidad Nacional Autónoma de México, 04510 Mexico City, Mexico; odetteodette@ciencias.unam.mx (L.O.G.-R.); emendozab@iingen.unam.mx (E.M.)
2 Instituto de Ecología, A.C., Antigua Carretera a Coatepec no. 351, Xalapa, 91073 Veracruz, Mexico; marisa.martinez@inecol.mx (M.L.M.); jorge.lopez.portillo@inecol.mx (J.L.-P.)
3 Unidad Académica de Sistemas Arrecifales-Puerto Morelos, Instituto de Ciencias del Mar y Limnología, Universidad Nacional Autónoma de México, 77580 Puerto Morelos, Mexico; vantuss@cmarl.unam.mx
* Correspondence: rsilvac@iingen.unam.mx

**Abstract:** The coastal zone is in a critical state worldwide, due to the loss and fragmentation of its ecosystems. Much of this is caused by long-term anthropic local, regional, or global actions, which drive coastal squeeze processes. Most of the criteria to evaluate the effects of coastal squeeze are focused on merely identifying its effect on the coastal zone. Here, we propose a framework to manage (identify, analyze, quantify, and tackle) the effect of coastal squeeze. This **DESCR** framework examines the relations between **Drivers**, **Exchanges**, and **States** of the environment to subsequently evaluate the chronic, negative **Consequences** and determine possible **Responses**. To illustrate the application of the DESCR framework, the coastal zone of Puerto Morelos, in the Mexican Caribbean, was studied using this approach. We analyzed the systemic interactions among the coastal ecosystems in this area, such as coral reefs, seagrass beds, beach, coastal dunes, and mangroves, which have been altered over the last decades, resulting in a severe coastal squeeze. Recommended responses include urgent measures for ecosystem management to mitigate the coastal squeeze.

**Keywords:** coastal ecosystems; climate change; Mexican Caribbean; long-term anthropic impacts; coastal squeeze

## 1. Introduction

The coastal zone is in a critical state in many parts of the world. The area covered by natural ecosystems is shrinking due to changes in land use, resulting in both loss and fragmentation of the area, aggravated by consequential changes in the natural processes caused by the anthropic modifications to physical, biological, and chemical drivers [1–4]. Together, these interacting drivers regulate the dynamic balance of coastal ecosystems [5].

In recent years, a more comprehensive vision on how the coasts function has been provided by studies from the perspective on how physical, biological, and chemical drivers interact with each other, and how the environment is modified by these drivers and their interactions. Understanding the relationships between fluxes of matter and energy and their consequences for coastal systems requires a trans-disciplinary approach, in which information derived from different areas of knowledge is integrated, e.g., ecology, geomorphology, geology, marine climate, socioeconomics, and legislation [6]. To date, there are studies in which the same physical processes are analyzed from different angles, for example, coastal flooding and erosion have been discussed in the specialized literature from an engineering perspective, focusing on the design and stability of structures and the potential effects of sea-level rise regarding the loss of land [7], whereas others have examined this loss from an ecosystem perspective (e.g., [5]). Moreover, other authors, e.g., [8,9], have addressed the consequences of these

processes from an economic perspective. Recent research revealed high levels of erosion on the world's sandy beaches (e.g., [10]), and the possibility of sandy beaches all but disappearing shortly [11], which requires that these processes be addressed by different disciplines as a whole, because the consequences are not limited to biophysical aspects. Several criteria have been proposed to face the challenges posed by sea-level rise so that coastal managers are encouraged to consider the concept of coastal squeeze (e.g., [12]).

Coastal squeeze has been defined as a process in which rising sea levels and other factors, such as hard infrastructure, cause loss of space in both directions—land and sea—and where the ecosystems no longer have the necessary conditions to maintain their essential functions (see Table A1—Appendix A). Most of the studies on this topic focus primarily on sea-level rise and hard infrastructure. A framework that provides a more comprehensive understanding of coastal squeeze and its driving factors can suggest strategies to assess the distribution and impact of this phenomenon and thus support management actions to deal with it.

Given the relevance and wide use of the term coastal squeeze, it is essential to have a common framework to evaluate it quantitatively to incorporate adaptive coastal management strategies and strengthen their practical application. Therefore, in this paper, we aim to improve the characterization of coastal squeeze and expand its definition based on multifactorial interacting elements. Although the term coastal squeeze has been used to describe other phenomena, such as the net effect of squeezing more people into smaller areas, we will only consider the coastal squeeze related to environmental impacts.

Here, we define coastal squeeze as local, regional, or global anthropic processes, inducing changes with long-term (chronic) negative consequences, which do not allow coastal ecosystems to adapt to global climate change, such as sea level rise and increasing storminess. To make coastal squeeze a useful and easily appliable concept for coastal management, it is important to develop criteria for its diagnosis, assessment, and decision making. The criteria should facilitate management based on accurate scientific evidence. The scheme proposed here is inspired by the Drivers-Pressure-State-Impact-Response (DPSIR) framework proposed by the European Environment Agency [13]. Patrício et al. [14] reported that since 1999, 25 schemes for management and decision-making across ecosystems have used the DPSIR conceptual framework derivations to structure and analyze information. The authors point out that even so, the framework has significant shortcomings. In particular, the assessment of pressures, state, and impacts oversimplifies environmental problems. The authors cited above recognize that clearer, more comprehensive, nested conceptual models are needed to quantify the links between pressure-state change in marine and coastal ecosystems.

The modifications made to the DPSIR framework here combine human requirements with natural processes. The factors considered in this modified framework are drivers, exchanges, state of the environment, consequences, and responses (DESCR) (Figure 1). The main advantage of the DESCR framework is that the natural and anthropic drivers and their associated bidirectional exchanges of fluxes of matter and energy with the environment are considered simultaneously. Unlike the original scheme, in which pressure causes unidirectional changes to the system [14], we consider the exchanges in which the drivers modify the state of the environment, and vice versa [6]. In the DESCR framework, instead of using the concept of impact, the term consequence is used to evaluate the long-term direct or indirect results of interactions between drivers and the environment [15], such as deaths, injuries, loss of property, impoverishment, disruption to economic activity, or environmental damage.

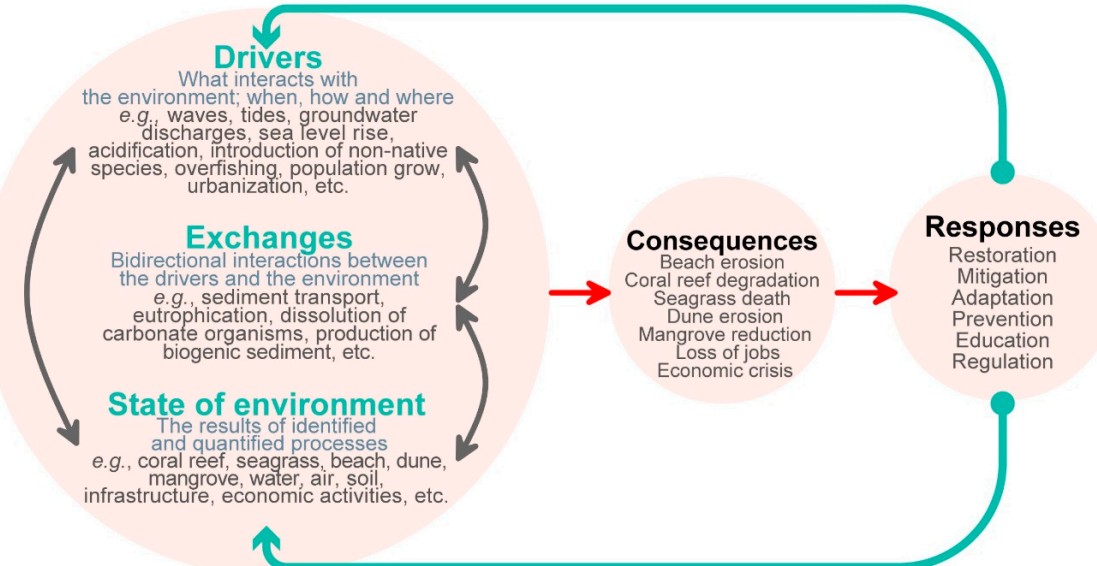

**Figure 1.** The DESCR (drivers, exchanges, state of the environment, consequences, and responses) framework (applied to a tropical coastal unit).

## 2. The Drivers–Exchanges–State of the Environment (DES) Cycle

In the first cycle of the DESCR framework, the dynamics and the elements affecting the ecosystems of the coastal unit must be examined. Then the environmental alterations that can potentially affect the intra- and inter-connectivity within the ecosystems must be identified. The next step is to determine the potential changes in ecosystem patterns and processes in terms of ecosystem loss or fragmentation, by the bidirectional exchanges between the drivers and the environment. It is crucial to bear in mind that changes in the coastal zone occur over large scales of time and space, i.e., from instantaneous to geological timeframes, and from local to regional or even global levels [6,16,17].

### 2.1. Drivers

A driver governs the direction of ecosystem change and could be of human or natural origin. The intensity, persistence, and frequency of some natural drivers may have increased or decreased due to human activities. To illustrate this concept, consider the sea waves as a driver. Waves are generated by wind-produced shear stress and are a type of oscillation of the free water surface. Wave characteristics depend on many variables (e.g., the intensity, persistence, and fetch of the wind, the sea bottom configuration, and the roughness of the seabed). Waves occur naturally in all coastal areas; however, the patterns of wave persistence, intensity, and direction are changing because of human activities. These changes are mainly associated with local infrastructure, dredging activities, sea-level rise, and the expansion of the tropics (e.g., changes in power, persistence, and direction of the waves [18]).

Drivers can be physical, such as non-local currents, tides, waves, winds, tsunamis, runoff, availability of light, temperature, dredging, and beach nourishment. They can also be chemical, such as pollution and acidification, or biological, such as organisms that modify the abiotic environment through their presence or biological activity. In so doing, biological drivers "maintain, modify, and create habitats" [19]. Such driving organisms can be considered "ecosystem engineers", either allogenic, when they induce large-scale changes in other adjacent ecosystems, or autogenic, inducing changes within their own ecosystem [20].

The temporal scales of the drivers can be short (e.g., hurricane), medium (e.g., ENSO), or long-term (e.g., sea-level rise). The assessment should be made at a local level, although the drivers can originate locally, regionally, or globally. As drivers are continually evolving, even without anthropic

interventions, the environment is always changing, and even ecosystems in pristine areas can be subject to coastal squeeze.

The consequences of coastal squeeze are assessed locally per coastal unit. However, the drivers are not necessarily of local origin. Local drivers that originate within a coastal unit can be controlled or regulated within the coastal unit (e.g., dredging, removal, and physical alteration of habitat, local pollution, introduction of non-native organisms, sediment extraction, pumping of water). Regional drivers are generated outside a coastal unit, but within its basin, i.e., the area bounded by the watershed of the hydrographic basin and the continental shelf. Some control can be exerted over them (e.g., underground and superficial runoff, waves, sediment supply, water quality, storm surge). On the other hand, global drivers are generally impossible to control at local level (e.g., tides and those associated with climate variability, climate change, eustatic movements, tectonic activity).

Parameters

The drivers may be assessed directly or indirectly, by constructing projections using available data. Given the random character and the uncertainty in predicting some of the drivers for coastal squeeze, their intensity, frequency, and persistence could be reported in terms of their probability of occurrence in a given return period.

For practical implementations, as proposed by [6], the anthropic or natural drivers can be divided into ecological (e.g., exotic species, groundwater and surface water nutrients, substrate/sediment quality); geomorphological (e.g., sediment features, sediment sources, local morphological evolution); geological (e.g., eustatic movements, tectonic activity); climatic (e.g., wind, runoff, wave, tides, currents, and sea level); socio-economic (e.g., population data, main economic activities, land use data, historical or cultural value of the area, population growth rate, disease), and legal (e.g., protected natural areas, protection measures, local planning). Each category could be parameterized by different indexes or measurable properties, which may also help to quantify and homogenize the information.

## 2.2. Exchanges

The analysis of exchanges determines how and to what extent the drivers interfere with the environment. These exchanges include the nutrient cycle, biological immigration, and emigration, species turnover, etc.

The spatial and temporal scales are key in considering exchanges. Given that the fluxes of mass and energy can occur on micro to macro scales, with different associated frequencies and intensities, any modification on any temporal or spatial scale may influence the intensity of the exchange and thereby produce a chain reaction in the environment.

The exchanges include sources, transport paths, and sinks. They can be physical (e.g., sediment transport, turbulence, energy dissipation), chemical (e.g., pH fluctuations induced by deviations of the carbonate-bicarbonate equilibrium, eutrophication), and biological (e.g., oxygenation of water through photosynthesis, carbon fixation, ecosystem processes induced by the activity of organisms). The exchanges are dynamic and can produce cyclical, episodic, or chronic effects. Exchanges can be evaluated through locally validated numerical models or parametrizations. Given the uncertainties in these exchanges, the probability of their occurrence should be included in the analysis to forecast potential scenarios.

## 2.3. State of the Environment

The state of the socio-environment concerns the properties of the space occupied by humans and ecosystems, be it a past, present, or hypothetical future scenario.

### 2.3.1. Spatial Delimitation: Which Coastal Ecosystems Should Be Included?

Although coastal squeeze affects many coastal systems, it has mostly been associated with inter-tidal habitats (salt marshes, mangroves), wetlands, and beaches (see Table A1—Appendix A).

However, other ecosystems that may be exposed to coastal squeeze include shallow sub-tidal and supra-tidal coastal ecosystems (e.g., [21–25]), ocean-land interfaces, as well as water bodies (hypersaline, saline/marine, brackish, or freshwater). Using these criteria, the ecosystems that should also be included in coastal squeeze analysis are seagrass meadows, macroalgal beds, mussel, coral and other types of reefs, sandy and rocky intertidal strips, estuaries, beaches, coastal dunes, coastal lagoons, coastal wetlands, mangroves, and salt marshes, as well as salt-intruded, aquatic, or semi-aquatic environments near the coast.

### 2.3.2. The Coastal Unit

The coastal unit is the space where coastal ecosystem exchanges occur, and includes shallow sub-tidal, inter-tidal, and supra-tidal zones. The boundaries of the coastal unit delineate the geographical area within which the coastal ecosystems interact and where the quantitative analysis of coastal squeeze takes place.

A coastal unit can be delimited for coastal squeeze analysis from a hydrological point of view, the watershed in terrestrial systems, or, from the coastal processes perspective, up to the continental shelf. Delimitation based on superficial hydrological basins is controversial, since these do not necessarily match the subterranean basins. A basin may be shared by more than one municipality, state, or even country. For practical purposes, it is more effective to use coastal units already established by regional coastal management programs. However, in doing so, the interdependence of processes of neighboring coastal units must not be overlooked, nor the possibility of redefining these units, when thought necessary. When delimiting a coastal unit, it is also essential to identify potential anthropic and natural obstructions that limit the ecosystems' spatial adaptation. Although set by rigid elements, the borders of a coastal unit must be considered open in the sense that fluxes or effects coming from outside can introduce changes inside the unit. A socio-ecological system involves considering the ecological and social parts of the environment as a single unit.

### 2.3.3. Parameters

The state of the environment is described through parameters that measure physical features (e.g., extension, turbidity, sea bottom configuration, cohesiveness, roughness); biological aspects (e.g., community structure, specific and functional diversity, presence of exotic species in each ecosystem within the coastal unit); chemical characteristics (e.g., nutrient concentrations, DBO, ODB, heavy metal concentrations); and man-made interventions (e.g., urban, aquacultural, or agricultural land use).

In probabilistic terms, it is necessary to evaluate how the state of the environment changes in response to the exchanges induced by the drivers. Modification of the drivers and exchanges must also be evaluated according to changes in the environment, as seen from the continuous updating and analysis of conditions that incorporate biophysical, economic, and social components.

## 3. The Consequence–Response (CR) Cycle

To formulate the responses of the framework, the consequences resulting from the DES cycle must be evaluated. The effects of one or more drivers are assessed, and the responses must tackle the origin of the altered exchanges, if possible.

The consequences are the qualitative and quantitative long-term effects on the patterns and processes of the socio-ecosystems induced by the DES cycle. One way to assess the consequences is through changes in coastal ecosystem services [26]. However, it would seem more practical to employ the concept of the receptor of the Source–Pathway–Receptor–Consequence framework [27] to assess the consequences. For example, the consequences of the loss of mangroves can be measured as economic (e.g., loss of employment, decrease in per capita income, damage to infrastructure due to flooding, etc.), human health (e.g., emergence of vector diseases), ecological (e.g., reduction in the number of species), physical (e.g., loss of land), political (e.g., changes in governance), cultural (e.g., loss of artistic heritage), etc.

The responses are the measures taken to reverse or mitigate coastal squeeze, which modify the state of the environment (restoration, relocation, etc.), or modify the activities of the community (regulation) or the attitudes of society (education and prevention). Decisions on the responses must balance human interests with preserving ecosystems, and may focus on conservation, prevention, recovery, mitigation, or adaptation actions. As examples of the successes and failures of the DESCR framework applications are made available, it is hoped that prevention combined with systematic, permanent monitoring would also be seen as a corollary action.

### 3.1. Consequences

The consequences may occur at different scales of time and space. Positive, negative, or neutral consequences on coastal ecosystems are possible. Negative, long-term consequences produce coastal squeeze when they are not part of a natural cycle and are not absorbed by the ecosystem.

In assessing coastal squeeze, the long-term consequences must be evaluated regardless of the duration or frequency of the exchanges that generate the modifications to the ecosystems. There may, indeed, be no consequences if the environment absorbs the impact of the drivers (e.g., tsunamis, hurricanes, construction). There is no coastal squeeze in this case. However, there could be transitory or permanent consequences. Again, there is no coastal squeeze when the ecosystem changes are cyclical or if these changes are of short duration, regardless of the intensity, duration, and cyclicity of the drivers. However, if the state of the environment is degraded in the long-term, then there is coastal squeeze, regardless of whether it is induced by episodic or cyclic drivers. The consequences are transitory when human interventions reverse a tendency of a coastal squeeze. However, some negative consequences are irreversible. Any natural consequence derived from the DES cycle that does not induce negative, permanent, or chronic alterations in the environment (loss of habitats/species) is not coastal squeeze.

### 3.2. Responses

The term response refers to actions taken to recover, prevent, mitigate, and adapt to coastal squeeze. These actions may include the implementation of green infrastructure [28–30] through ecosystem-based management strategies [6,31,32] and policies that improve the consequences of the DES cycle.

Responses can be structural (e.g., construction, restoration, or relocation of coastal infrastructure) or non-structural (e.g., policies, education, protection, conservation). The first aim to control risk, the latter aim to avoid risk. Structural responses focus on the protection of human interests, while the non-structural responses focus on adaptation. The responses should be part of Integrated Coastal Zone Management (ICZM) and Disaster Risk Reduction (DRR) programs. Infrastructure development (roads, commercial areas, residential housing, etc.) may be addressed by ICZM, while DRR influences investments in coastal protection structures that may inadvertently cause coastal squeeze.

The response should also promote nature-based approaches [6,30,32,33] and include implementing educational programs and enforceable regulatory frameworks. Long-term coastal problems are frequently detonated by lack of reliable data [6], insufficient training of decision-makers, the rejection of management plans by the community concerned, failure to comply with environmental laws, corruption, economic pressures, and insufficient funding for maintenance and monitoring programs [5]. Responses may have unintended and potentially harmful side effects if we do not have a comprehensive framework to assess coastal squeeze.

## 4. An Example of Coastal Squeeze Assessment

To illustrate the proposed methodology, the coastal unit of Puerto Morelos, Mexico, is presented as a case study (Figure 2). This unit corresponds approximately to the coastal area defined in the local coastal management program. The coastal unit is limited in the north by Punta Nizuc and in the south by the commercial port of Puerto Morelos. The Cancun—Puerto Morelos highway, located landwards, is the western limit, and the eastern site is delimited by the continental shelf, which slopes abruptly downwards at approximately 20 m depth (Figure 3b).

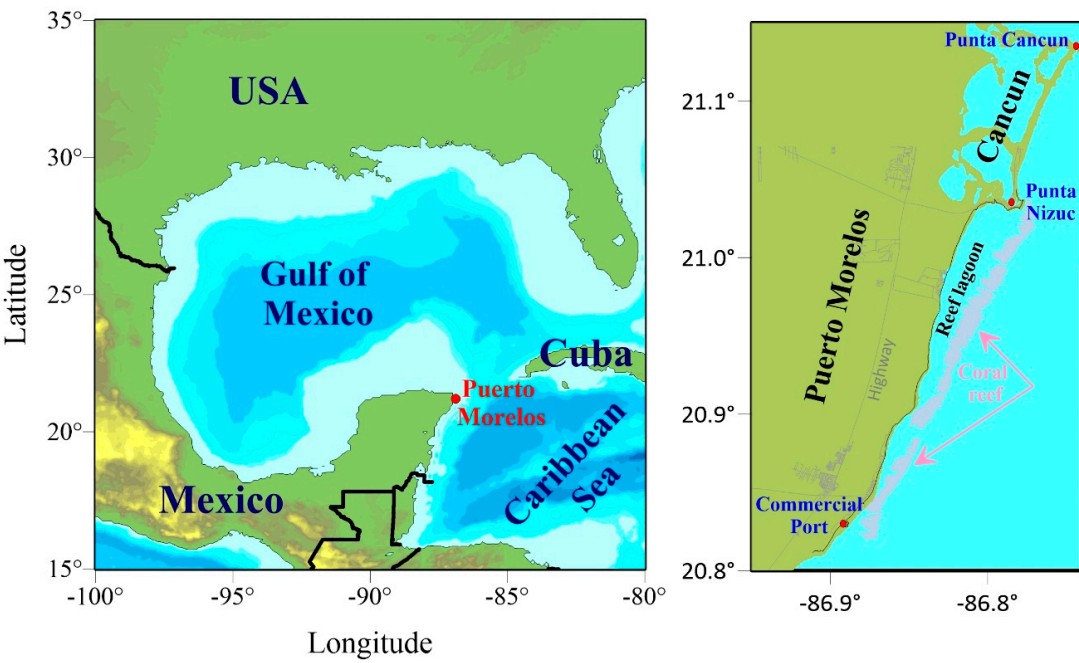

**Figure 2.** Location of Puerto Morelos, in the state of Quintana Roo, Mexico. The image on the right indicates coral reefs distribution and the reef lagoon, often occupied by seagrasses.

*4.1. The DES Cycle (the State of the Environment, Exchanges, and Drivers)*

Puerto Morelos (20°50′53.6964″ N, 86°52′33.9816″ W) is located 36 km south of Cancun, on the Mexican Caribbean. The area is characterized by shallow relief and the absence of surface rivers due to the limestone geology. The town of Puerto Morelos is, on average, 3 to 5 m above sea level. The main economic activity of Puerto Morelos is tourism, which is expected to continue to increase in the future.

Maps with relevant information of the coastal unit (Figure 3a,b) were constructed based on data from [9,34,35]. Ecosystems in the coastal unit include coral reefs, seagrass meadows, beaches of fine biogenic sand, vegetated small sand dunes, and mangroves (Figure 3a). A fringing reef, running parallel to the coastline, forms part of the Mesoamerican Reef System. In addition to its ecological value, the reef provides important protection services (e.g., against the action of the waves, preventing coastal erosion, and offering safer breeding and feeding areas for invertebrates, sea turtles, and fish). Even so, there are some eroded stretches of the beach. The reef flat is a narrow platform 1 to 3 m deep and 50 to 200 m wide (Figure 3a,b). It is mostly covered by scleractinian corals [34]. The reef is 0.5 to 1.5 m deep and 30 to 100 m wide, with four very distinct areas: the fore reef (seawards), reef crest, back reef (landwards), and the reef flat zone in the lagoon [36,37]. Towards the ocean, beyond the coral zone, the seabed slopes very steeply, and biophysical process are controlled by oceanographic mechanisms. Between the reef and the coast lies a 350–1600 m wide and 2–6 m deep lagoon (Figure 2), mostly covered with seagrass meadows, with patches of exposed limestone with soft coral communities. The beach and the dunes are composed of fine white sand, with a very gentle slope [38,39]. The mangroves are separated from the sea by a sand barrier of primary (embryo) dunes vegetated with native and exotic plant species and is fragmented by urban and tourist infrastructure. The coverage of wetland has been severely diminished by felling, drying, and the modification of water flows, usually to make way for the construction of tourist infrastructure and roads. As a means of illustrating the main drivers, their biophysical exchanges and possible response in the state of the mangrove, dune, beach, seagrasses, and coral, additional information on the main characteristics of these ecosystems appears in Table A2 (Appendix A). It is important to note that changes in the state of the environment will eventually generate different exchanges and characteristics of the drivers. Similarly, with different driver intensities and frequencies, the exchanges, and therefore the state of the environment, will be different.

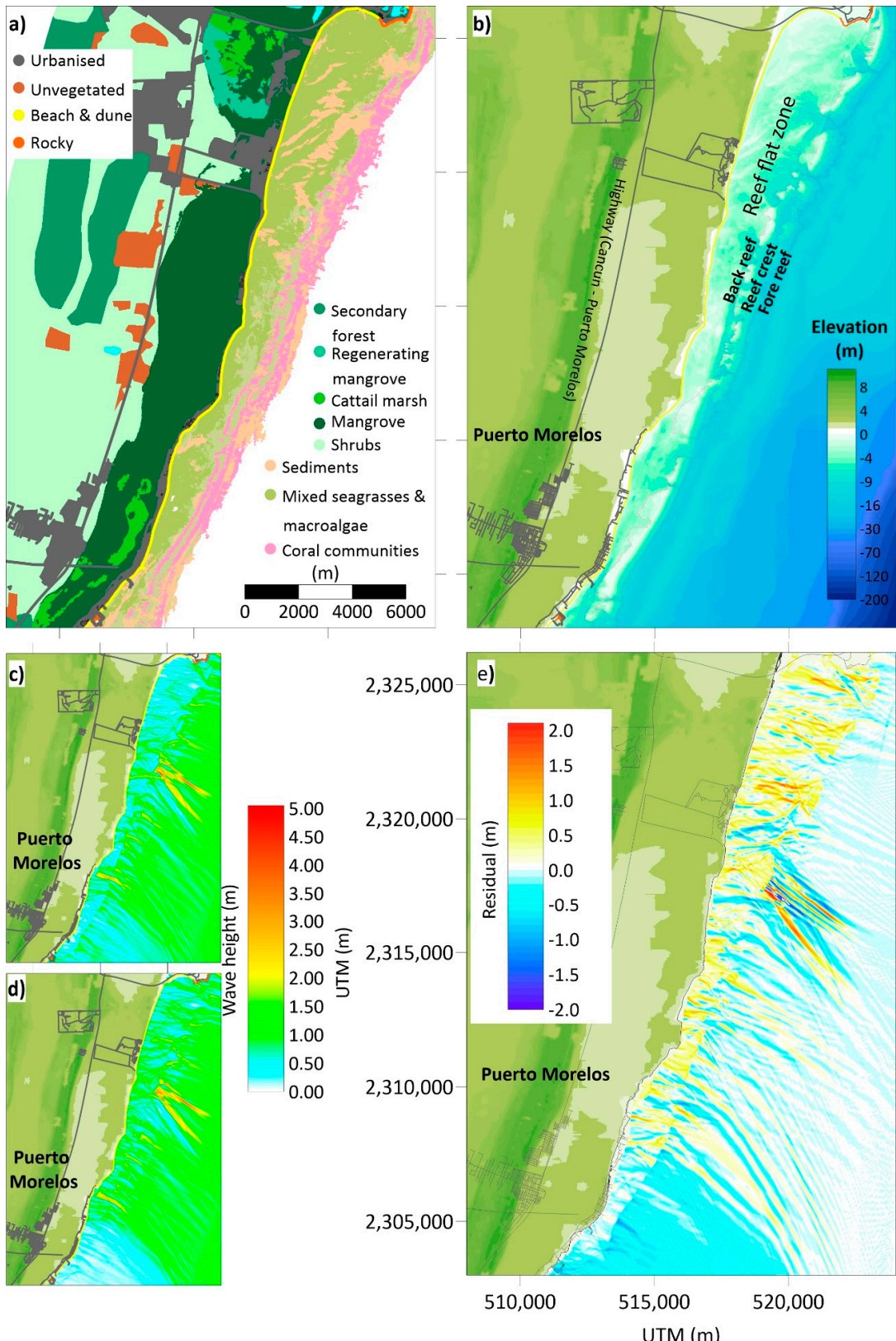

**Figure 3.** (**a**) Coastal ecosystems and urbanization, (**b**) T\topo-bathymetry and roads, (**c**) maximum wave height under present conditions, (**d**) maximum wave height map under sea-level rise (+0.5 m) and absence of coral reef and seagrass, (**e**) local changes in wave height between the present sea level scenario and a sea-level rise of 0.5 m (colors from yellow to red imply an increase in wave height, while blue implies a decrease).

Part of the urban area lies on the sand-barrier, but most is further inland, beyond the 2 km wide mangrove forest. There is no centralized sewage system for the urbanized areas, and wastewater (without retrieval of nutrients) is discharged into the aquifer. Due to the limestone (karstic) geology and the gentle slope of the land (Figure 3b), this wastewater travels towards the sea through underground flows, some of which emerge in the mangrove area, the reef lagoon, and into the sea beyond the reefs, through submarine springs or fissures.

The main exchanges in the coastal unit are associated with (1) the hydrodynamics induced by groundwater runoff, waves, tides, storm surge, coastal currents, and winds; (2) sediment transport by wind and water currents; (3) photosynthesis; (4) nutrient input and recycling within the ecosystems; (5) inter and intraspecific species interactions; (6) biological migration and species turnover; (7) ecosystem modifications induced by key species; (8) the dissolution of carbonate organisms; and (9) the production of biogenic sediment.

Waves govern the physical processes in the reef lagoon [40]. The waves are more intense in winter; however, the most energetic storms are generated by sporadic tropical storms and hurricanes [41]. According to previous studies [42,43], most wave energy dissipates on the reef crest, due to wave breaking, friction, and turbulence in the shallow water. Behind the crest, seagrasses play a significant role in reducing the speeds of the coastal currents, filtering suspended sediment and producing and retaining sediment between the roots and rhizomes.

Until 2015, before recurrent massive influxes of pelagic *Sargassum* species (*Sargassum natans* and *S. fluitans*) affected the area, the seagrass meadows were very stable [44,45]. Since autumn 2015, there have been chronic changes in the seabed configuration with mortality of nearshore seagrasses, which have subsequently been replaced by algae [30]. Based on data provided by nine hotels, [46] reported that 2.4 m$^3$/m (in 2015), 3.2 m$^3$/m (in 2018), and 1.7 m$^3$/m (in 2019) of sargasso were removed per month, on average, from beaches in Puerto Morelos and la Riviera Maya. The presence of the sargasso and its removal to nearby dumps severely affected the coastal ecosystems and tourism activities. Severe degradation of water quality, caused by the lixiviates and organic particles of sargassum, affects the whole lagoon system [46]. Loss of live coral [47], attributed to a disease (white syndrome) [47], aggravated by a decline in water quality has reduced the rugosity of the coral and a flattening of the reef crest, lessening wave energy dissipation through turbulence, friction, and wave-breaking [48].

To illustrate the importance of the sea-level rise and the health of some ecosystems, Figure 3c,d show the maximum wave height for the same wave conditions. However, as Figure 3d shows a sea level increased by 0.5 m, consequently, the roughness of the seagrass and coral reefs were not considered. The scenarios were obtained using the wave propagation model WAPO [49]. This model uses a second-order approximation of the modified version of the mild-slope and can implicitly solve refraction, reflection, shoaling, diffraction, and dissipation by friction and wave breaking. The model was forced with a long swell induced by Hurricane Matthew, 1 October, 2016 (wave height Hs = 1 m, peak period Tp = 15 s, and incident direction θ = 120° clockwise from the north; [50]). The wave features and intensities are very similar in the two scenarios from the open sea to the fore reef zone. However, from the reef crest to the shoreline, the wave conditions are not similar. Figure 3e shows the local residuals in wave height between the current sea level scenario and considering a sea level rise of 0.5 m in detail. Changes in wave energy patterns, associated with rising sea levels and the reduction of turbulence and friction, induce modifications in local current patterns and sediment and nutrient transport, causing direct or indirect reshaping of all the coastal ecosystems and adjustments to the state of the environment.

Figure 4 outlines four scenarios that illustrate (a) the hydrodynamic conditions before the massive arrivals of sargasso from 2015, (b) the present condition, (c) the potential condition if the processes maintain the same trends (the do-nothing response), and (d) the hypothetical condition associated with mitigation and adaptation measures.

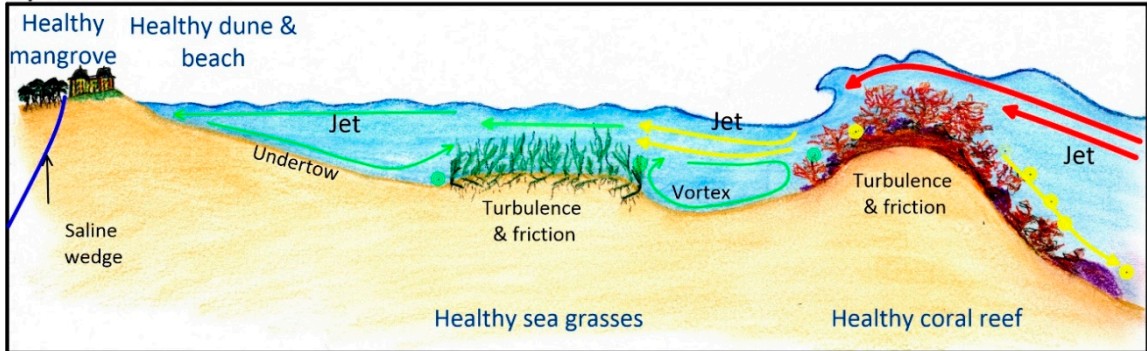

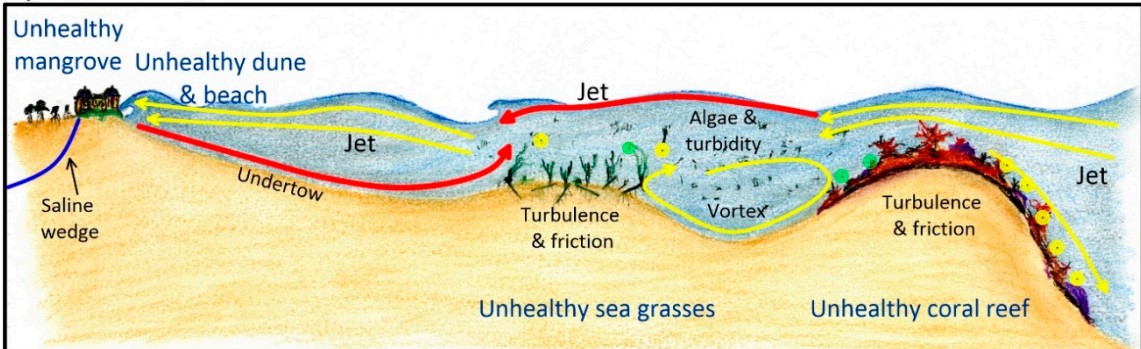

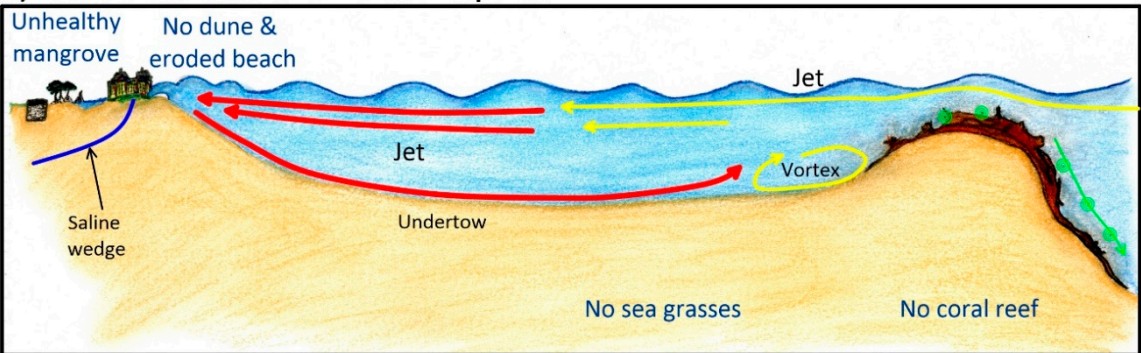

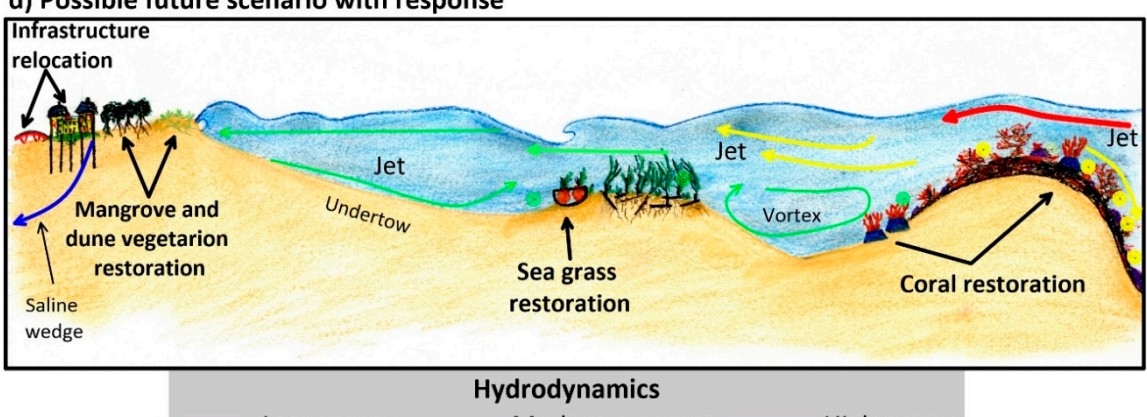

**Figure 4.** Schematic scenarios of a cross-section of the Puerto Morelos coastal unit: (**a**) scenario before 2015, with reasonably healthy ecosystems; (**b**) present conditions with some degradation of ecosystems; (**c**) future scenario with the do-nothing response; and (**d**) future scenario with responses of mitigation and adaptation implemented.

When the waves hit the coral reef, the effects of turbulence and friction inside the reef crevices, and wave breaking above the reef crest, induce considerable energy dissipation (Figure 4a). A small proportion of the wave energy reflects seawards, and the rest is transmitted to the beach. The coral reef is composed of species with a high degree of roughness, such as *Acropora palmata*, which grow on the reef crest. Meanwhile, the leeward face is populated by species less resistant to wave breaking (*Pseudodiploria* ssp. and *Orbicella* spp.). The turbulence within the reef interstices and wave breaking generate a high oxygen concentration in the water, which, together with low turbidity levels, provides ideal conditions for certain species of marine fauna and flora, such as seagrass. Coral reefs are also significant producers of biogenic sand, which feeds the beach and the dunes.

Seagrass beds are found landward of the reef, in shallow water, and on sediment (coarse sand to clayey silt). The seagrass leaves (or canopy) dissipate wave energy through friction, thereby allowing fine particles in the water to settle on the seabed. This accumulating sediment is retained in the dense root-rhizome mats of the seagrasses, a habitat for many species of fauna.

On the leeward side of the seagrass beds is an area with low energy waves. The beach has a very gentle slope (Figure 4a) and is made up of fine sand particles. Since there is little wave energy and the tidal range is small, wind-transported sediment is limited, and dunes are low. Since the water table is shallow, there is abundant vegetation, and this becomes a natural barrier that prevents the sediment from moving inland. Mangroves are found landward from the dunes, giving further protection to inland areas by their extension and complex structure (Figures 3a and 4a).

Based on the data reported by [46], various scenarios were constructed. Figure 4b outlines the prevailing conditions after the first sargasso brown tides. Persistent physicochemical changes in the water quality (e.g., temperature, light, pollution, and acidification) affect the whole system, probably including the coral reefs [47]. With even a small rise in the sea level, key species of hard coral die, and reef roughness is lost. Loss of roughness means that less energy is dissipated, more energy is transmitted to the coast, and less oxygenation occurs in the water.

As the conditions change, some populations responsible for sand production decrease, such as parrotfish [51]. On the other hand, some calcareous algae may appear [52]. The amount of biogenic sediments from the coral reef will probably diminish overall, and the beach will have a sand deficit [9].

There is more wave energy between the coral reefs and the beach. The sediment retained by seagrasses in the root-rhizome mats is now transported along and across the coast and lost. With less stable substrata, the seagrass beds decrease in extension and become fragmented. As there is now less friction due to seagrass loss, more wave energy reaches the beach.

As the wave energy increases, part of the fine sediment is transported from the submerged beach landwards, which allows the wind to build higher dunes. Because the waves reflected by the beach and coastal protection structures also have more energy, they transport large quantities of sediment seaward, eroding the beach. Turbidity increases, because the transport of fine sediment is more significant than the filtration and retention by seagrasses, accelerating the death of many coral species on the reef and triggering the replacement of seagrasses by algae. Without seagrasses, more wave energy reaches the reef lagoon and the beach.

In line with the research carried out by [9,53], this cyclical process is repeated until the corals die. The seagrass meadows disappear, and the beach reaches a new equilibrium (Figure 4c), with a steeper slope of coarser sand and higher dunes. Coastal infrastructure may be damaged. The rise in sea level causes a decrease in the beach and dune filtering effect, inducing saline intrusion landwards, and flooding lower areas of the coastline. These consequences eventually modify the quantity and diversity of the mangrove ecosystem behind the dunes.

Comparing Figure 4a,c, or Figure 5a,b, there are five types of coastal squeeze in this coastal unit induced by physicochemical conditions on the coral reef: (1) undertow and sediment transport on the seagrass beds, (2) undertow and sediment transport to the beach, (3) sediment transport and the infrastructure on the dunes, (4) sea-level rise and saline wedge advance, and (5) infrastructure on the mangroves.

A schematic summary of the exchanges is presented in Figure 5. The equilibrium condition should not be interpreted as a static condition, since any ecosystem needs to grow, mature, and regenerate. Dynamic equilibrium is achieved when the exchanges of matter and energy associated with periods of rain and low water, and of periods of calm, as well as exposure to storms (such as hurricanes) give stable conditions.

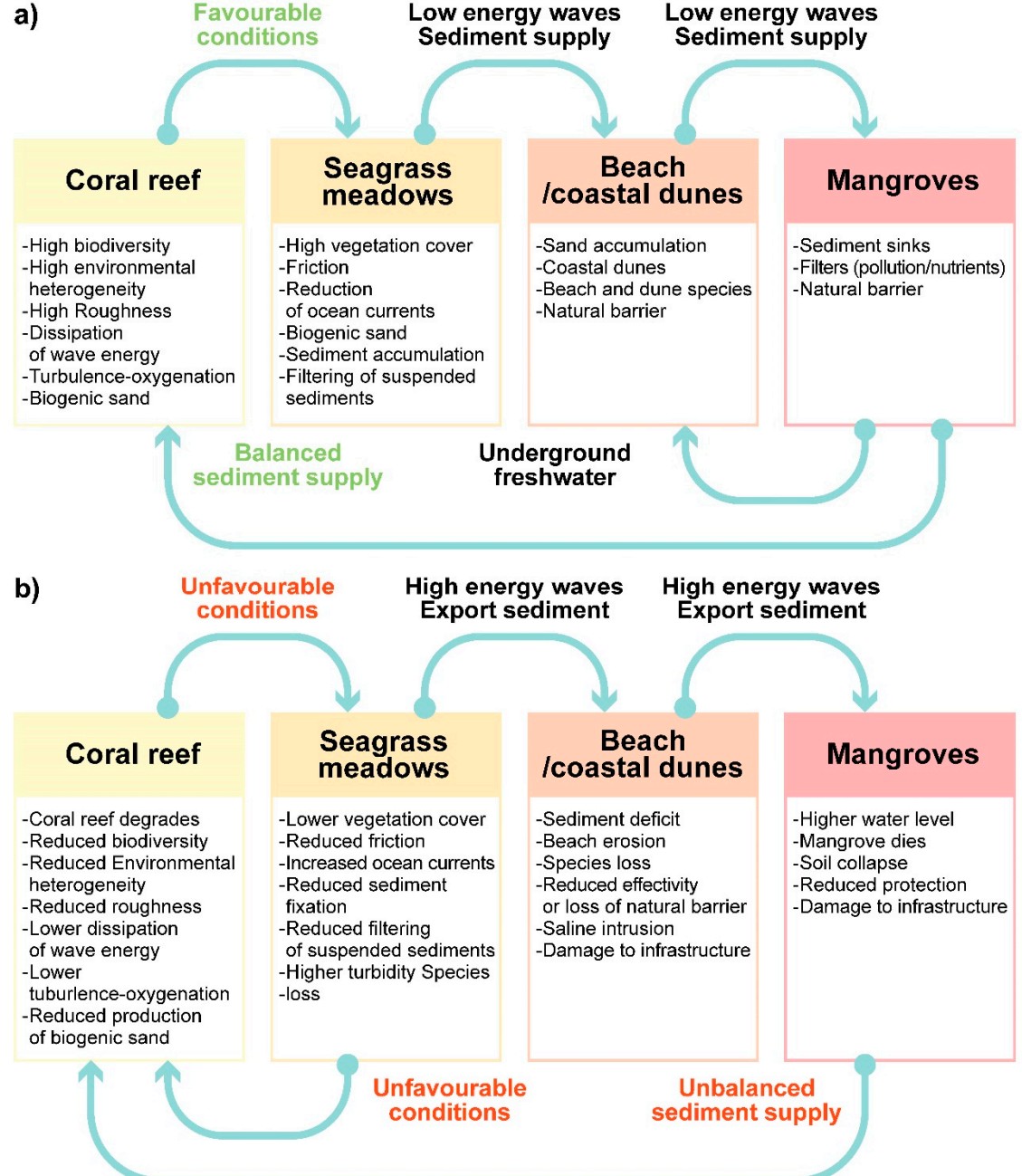

**Figure 5.** (**a**) Equilibrium conditions—without coastal squeeze; and (**b**) altered dynamics (sea-level rise + low water quality)—produces coastal squeeze. Arrows indicate the exchanges.

### 4.2. The CR Cycle (Consequences and Responses)

Changes in coral reef health associated with the rise in sea-level and water quality degradation have far-reaching consequences, such as coastal squeeze, because of the interconnectivity among the coastal ecosystems (coral reefs, seagrass meadows, beach, coastal dunes, and mangroves). Unfortunately,

countering the global effects of climate change, such as sea-level rise, is almost impossible. However, there is room for maneuvering at regional and local levels, by focusing responses on the following aspects:

- Executing integral coastal management strategies (regulating activities and use of the space). Over recent decades, the lack of adequate environmental planning for Puerto Morelos has been evident in frequent changes in policy and financial support.
- Implementing mitigation and management plans to deal with the original pressure (in this case, the sargasso influxes) and establishing strategies to respond rapidly and effectively to such emerging problems.
- Restoring seagrasses. By starting with fast-growing and tolerant pioneer species (e.g., *Halodule wrightii*) in nearshore areas, seagrass restoration will reduce the onshore undertow effect and avoid the resuspension of sediment between the rhizomes of the seagrass beds. Special precautions must be taken with these species, since they do not generate deeply-rooted root-rhizome mats and are, therefore, very easily removed by the action of waves or currents. Therefore, these efforts should be followed-up by enhancing the colonization of more robust and deeper-rooted seagrass species, such as *Thalassia testudinum*.
- Restoring corals. The first step is to reduce wave dissipation by increasing turbulence and bottom friction effects. These functions can be temporarily mimicked with artificial structures and other restoration techniques, such as coral species transplantation. The increased turbulence will generate more water oxygenation, improving colonization opportunities for back-reef species.
- Restoring dune vegetation. When the beach's wave energy is less, the slope will be gentle, and the dune accretion process will return. It may be necessary to accelerate this process by nourishing the dune and the beach, recycling local sand from the sea bottom. To stabilize the dune, vegetation will be required. The aerial parts of the plants will control the wind erosion of the dune.
- Restoring mangroves and other wetlands. If the level of water in the mangrove increases or stagnates, channels may be excavated, and the surface level elevated to facilitate water flow to the sea [54,55].
- Improving wastewater treatment. By mitigating the consequences of the poor wastewater system treatment, coral and seagrass degradation will be less. By restoring the groundwater conditions in the subsoil, mangrove species will be less likely to be replaced by cattail marsh.
- Altering infrastructure. Some infrastructure (e.g., buildings, roads) was designed with no thought for environmental sustainability. Relocating it landward may be the only way in which essential ecosystems can be restored. In other cases, infrastructure can be adapted so that it stands on stilts or pillars. The use of continuous foundations must be avoided so that superficial and sub-surface flows are not interrupted. Roads must have sufficient water passes, so that excess water can drain off and flow away, to ensure different water catchments remain connected. It is essential to guarantee mangrove health and its adaptation to new water level conditions.
- Reducing socioeconomic inequality. By encouraging other types of economic activities, the pressure on the coastal ecosystems will lessen, and economic vulnerability will become less acute due to the variability in the influx of tourists.
- Controlling migration into the area. The population of Puerto Morelos tripled from 2010 to 2015, from 10,000 to almost 30,000 inhabitants. Proper, long-term planning will allow the authorities to provide the services and infrastructure the inhabitants deserve.
- Implementing educational programs. The production of information about the ecosystems in Puerto Morelos, easily understandable and freely available, will strengthen the local community's environmental responsibilities. This type of action will help temper the lack of socio-environmental awareness, respect for laws, and corruption at various levels of society and government.
- Monitoring the coastal unit. Continuous monitoring of the environment will produce reliable information that must be made public, reduce uncertainties, and ensure that appropriate actions are taken to adapt measures when necessary.

## 5. Discussion and Conclusions

The concept of "coastal squeeze" has penetrated significantly into the decision making of coastal managers and those concerned with reducing disaster risk, e.g., [56]. Therefore "coastal squeeze" assessment techniques should be disseminated by the scientific community to develop policies that encourage better use of the coastal areas and improve the balance between natural processes and human interests/needs.

As recognized by [14], despite the polishing done to the DPSIR framework, there are still some aspects that need to be improved for its practical application, particularly as it does not allow for the incorporation of complex interactions. The DESCR framework presented here employs the practical concept of the coastal unit, that includes all interacting ecosystems within a geographical setting. Furthermore, the DESCR considers the bidirectional exchanges of matter and energy between the Drivers and the State of the environment, replacing the concept of pressure by exchanges. We consider that the DESCR may provide a more practical framework for the study of interacting ecosystems than the DPSIR for the study of coastal squeeze.

The coastal zone is the most dynamic environment on Earth, the only region where the terrestrial environment, the atmosphere, seawater, and freshwater interact at different scales of intensity, time, and space. The coastal zone is transformed as it absorbs energy from the sun, winds, tides, waves, runoff, and groundwater discharges. It is also affected by, and affects, the processes and flows that occur on the coast and modify substrate and water properties [57]. These processes induce positive and negative feedback on the dynamics of coastal ecosystems [6]. However, anthropic stresses can produce irreversible negative alterations in coastal ecosystems.

Coastal ecosystems are very vulnerable to exchanges of matter and energy produced by drivers. The intra- and interconnections between them are sensitive to modifications of the environment and drivers. The persistence, frequency, or intensity of drivers can bring about deterioration in an ecosystem, affecting processes in the neighboring ecosystems, which may also modify adjacent ecosystems in a cascade effect. When ecosystem health and connectivity thresholds are exceeded (often unexpectedly), a point of no-return may be reached, with irreversible consequences.

Continuous generation of, and access to, information, and a well-informed, critical community are essential to understand and make timely responses to scenarios of coastal squeeze. While many predictions estimate climate change variation to be gradual, these changes may be relatively abrupt and unexpected (e.g., [58]). For instance, from data concerning the last interglacial period in the Puerto Morelos region, "The buried reefs revealed that sea level rises of as much as five centimeters per year resulted in at least a two meter jump in as little as 50 years, based on a series of reefs retreating closer to a receding shore over time [59]".

Quantifying coastal squeeze through the DESCR framework can be integrated into existing methodologies, such as the source–pathway–receptor–consequences [15,60]. This would have the additional advantages of explicitly considering uncertainties from diverse sources and improving risk management.

The present work aims to bridge the communication barrier between technical and non-technical decision-makers. The DESCR framework can be used to determine the threshold between sustainable and non-sustainable coastal uses and set priorities in investment, monitoring, and management programs. It is hoped that these results will help us move to a more sustainable use of coastal resources.

**Author Contributions:** Conceptualization: R.S.; formal analysis: R.S., M.L.M., B.I.v.T., L.O.G.-R., E.M., and J.L.-P.; investigation: R.S., M.L.M., B.I.v.T., L.O.G.-R., E.M., and J.L.-P.; resources: R.S., M.L.M., B.I.v.T., L.O.G.-R., E.M., and J.L.-P.; data curation: R.S., M.L.M., B.I.v.T., L.O.G.-R., E.M., and J.L.-P.; writing—original draft preparation: R.S., M.L.M., and L.O.G.-R.; writing—review and editing: R.S., M.L.M., B.I.v.T., L.O.G.-R., E.M., and J.L.-P.; visualization: R.S., M.L.M., B.I.v.T., L.O.G.-R., E.M., and J.L.-P.; supervision: R.S.; project administration: R.S., and M.L.M.; funding acquisition: R.S. All authors have read and agreed to the published version of the manuscript.

**Funding:** This research was funded by the CONACYT-SENER-Sustentabilidad Energética project: FSE-2014-06-249795 Centro Mexicano de Innovación en Energía del Océano (CEMIE-Océano).

**Acknowledgments:** The authors wish to thank the CEMIE-Océano (project 249795). We are grateful to Guadalupe Barba Santos (Seagrass lab, UNAM-ICML), Edgar Escalante, and Miguel Gómez (SAMMO, UNAM ICML) for providing invaluable information.

**Conflicts of Interest:** The authors declare no conflict of interest.

## Appendix A

**Table A1.** Previous definitions of Coastal Squeeze.

| Definition | Ecosystem/Perspective | Main Causes |
|---|---|---|
| Anthropogenic barriers prevent wetlands from migrating inland, and steep slopes bordering wetlands stall or completely halt wetland migration [61]. | Wetlands | Hard infrastructure |
| Related to coastal steepening or narrowing, the process whereby the cross-shore profile does not retreat or advance [62]. | Cross-shore profile | Sea level rise |
| Coastal habitats and natural features are progressively lost or drowned, caught between coastal defenses and rising sea levels [63]. | Coastal habitats | Sea level rise Hard infrastructure |
| The process where rising sea levels and other factors such as increased storminess push the coastal habitats landward [64,65]. | Coastal habitats | Sea level rise |
| Coastal habitats are progressively reduced in area and lose functionality when caught between a rising sea level and fixed sea defenses or high ground [66]. In this case, there is a loss of the intertidal area [67] and habitats [68]. However, coastal squeeze does not refer to losses due to natural processes [56]. In many estuarine environments, flood and coastal defenses constrain saltmarshes producing losses in intertidal habitat [21]. | Intertidal habitats | Sea level rise Hard infrastructure |
| Sea level rise coupled with shoreline armoring creates a "coastal squeeze" of habitat loss from both directions for many narrow beaches that no longer have an adjacent upland area for subsidence or retreat [69]. | Beach | Sea level rise Hard infrastructure |

**Table A2.** Important features of the main coastal ecosystems in the Puerto Morelos coastal unit. Data sources [35–37,45,70–74].

| | Drivers | Exchanges | State |
|---|---|---|---|
| Mangrove | Wind Runoff (superficial and groundwater) Salt intrusion Nutrient and pollutant discharge Compaction Bio-modellers (roots, wood, crabs) | Flood control $O_2$ production Organic matter and nutrient source Carbon sink Regulation of freshwater discharge Soil formation Soil retention Water table oscillations Wind intensity reduction | Land-locked mangrove forests dominated by red mangrove—*Rhizophora mangle* (high flood patterns and moderate salinity) White mangrove- *Laguncularia racemosa* (moderate flood patterns and low salinity) Black mangrove-*Avicennia germinans* (infrequent flood patterns and high salinity) Extensive sawgrass (*Cladium jamaicense*) patches combined with spikerush *Eleocharis cellulose* and cattail *Typha domingensis*. Main, paved, and dirt roads have fragmented the original mangrove ecosystem. Golf courses have replaced mangrove areas. |

**Table A2.** *Cont.*

| | Drivers | Exchanges | State |
|---|---|---|---|
| **Dune** | Wind<br>Runoff—sub superficial<br>Sea level<br>Sediments<br>Nutrients<br>Salt intrusion<br>Water table fluctuations<br>Bio-modellers (roots, wood, crabs) | Carbon sink<br>$O_2$ production<br>Organic matter and nutrient production<br>Regulation of saline intrusion<br>Sand stabilization<br>Soil formation<br>Water table oscillations<br>Wind intensity reduction | Windward- *Sporobolus virginicus, Sesuvium portulacastrum*<br>Dune crest—*Croton punctatus, Tournefortia gnaphalodes, Suriana maritima, Scaevola plumerii, Chrysobalanus icaco*<br>Leeward—*Thrinax radiata, Coccothrinax readii, Pseudophoenix sargentii, Caesalpinia vesicaria, Pithecellobium keyense, Bravaisia berlandieriana, Coccoloba uvifera, Cordia sebestena, Metopium brownei*<br>At the north and south of the coastal unit, the coastal dune has been replaced by tourist and housing infrastructure. |
| **Beach** | Wind<br>Sea level<br>Tides<br>Waves<br>Currents<br>Sediments<br>Bio-modellers (turtles, crabs) | Wave/current energy dissipation<br>Sea turtle nesting<br>Sediment transport | Healthy corals and vast seagrass meadows, and fine white sand (>0.3 mm) with a very gentle slope. Wave energy dissipates as it crosses the reef and seagrasses. The wind-generated waves in the reef lagoon are dissipated by a spilling break, with a small swash zone. The energy of the reflected waves is small.<br>In some areas, beach erosion and instability have been reported.<br>Around a dozen jetties and groynes are found along the beachfront of the unit. |
| **Seagrasses** | Tides<br>Waves<br>Currents<br>Sediments<br>Nutrients<br>Pollutants<br>Bio-modelers (turtles, fishes) | Sediment retention<br>Sediment production<br>Organic matter and nutrient production<br>Exportation of mature fish<br>Turbidity reduction<br>$O_2$ production<br>Carbon sink | Extensive submarine meadows covering the bottom of the reef lagoon composed of *Thalassia testudinum* (robust climax species under stable conditions), *Syringodium filiforme* (dominates in areas with higher nutrient concentrations) and *Halodule wrightii* (dominates in disturbed areas), accompanied by calcareous sand-producing rhizophytic algae.<br>Water quality in the areas is affected by the submarine springs containing wastewater.<br>The *Sargassum* influxes also affects the process of photosynthesis. |
| **Coral** | Tides<br>Waves<br>Currents<br>Sediments<br>Nutrients<br>Acidification<br>Pollutants<br>Bio-modelers (algae, sponges, polychaetes, urchins, fishes) | Wave/current energy dissipation<br>Fish and invertebrate growth and production<br>Organic matter and nutrient production<br>Sediment production<br>$O_2$ production<br>Carbon sink | Optimal development in highly hydrodynamic areas in warm (22–28° C), clear waters. There is a relatively large coral cover on the reef crest and a lesser cover of *Acropora palmata* in the shallower sector of the reef crest, while *Orbicella* spp. and *Pseudodiploria* sp. dominate the back reef. Other benthos includes sponges and calcareous and fleshy algae.<br>The reef status is assessed through indicators such as the abundance of key organisms, density and abundance of reef-building corals, the abundance of fishes, and macroalgae abundance. A wide range of statuses (from healthy to stressed) has been reported. |

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
