# Peer review of "A Framework to Manage Coastal Squeeze"

_sustainability, doi:10.3390/su122410610_

Round 1

Reviewer 1 Report

Paper Summary:

In this paper, the authors present a framework to access coastal squeeze in systems along the coast. Coastal squeeze of ecosystems is a very important topic for coastal communities faced with anthropogenic pressures and sea level rise. Formulating a framework that can be used by both technical and non-technical audiences is an ambitious and significant goal. Through a case study, the authors explain how their framework can be used to detail the components of coastal squeeze and future scenarios of the coastline.

General Comments:

I have three main comments, but the first is the most important to deal with. The paper suffers from a lack of clarity and organization. Examples of this include: 1) over use of bulleted lists that are long and do not highlight important parts; 2) Table 2 is confusing in both format and what points it is trying to get across; and 3) Case study description needs to be streamlined to the important parts of the system for coastal squeeze. These are just a few examples, but I think the authors need to decide on the important points of the paper and try to trim back the rest.

My second comment is that the authors should detail how to use the framework quantitatively. At the moment, it is just description, except for the very short “Parameters” sections. Details on how to apply this system quantitatively is critical for the adoption of this framework by other users. Lastly, the figures really need improvement. They need to be higher resolution and the text needs to be bigger. See specific comments below, but it was hard for me to even understand the figures because I could not read them. Please see below for a list of specific comments on the document.

Specific Comments:

Stylistic comment – Watch for use of passive voice. I also find there are too many bulleted lists. It is a good way to highlight an important list – but is overused in this paper.

Title – Why is “Coastal Squeeze” capitalized in the title but nowhere else?

Section 2.1.3 – I find the use of the word extension here confusing. The authors should clarify what they mean by this section.

Sections 2.1.4, 2.3.3 – I find the parameters section short and somewhat confusing. I think they authors could spend more space explaining how to estimate drivers if this framework is supposed to be quantitative. I think that these parameters sections could be merged into a larger section about quantitatively analyzing these components of the system.

Lines 268-293: This site description section could be shortened and refined to only discuss targeted information. It is too long at the moment.

Table 2 – Table 2 is really confusing and hard to follow. The cells seem to be lists or paragraphs with no clear organization. Please distill into the most important points and condense to a half page.

Figure 1 – the quality of the figure and the colors chosen make it almost impossible to read text. Please convert to higher quality figure file and change the light blue and yellow combination.

Figure 3 – Hard to differentiate the changes in wave climate between panels c and d. Is there a way to highlight it better? Also, in panel a, the legend should all be within the panel rather than spanning across multiple panels at the top.

Figure 5 – Size of text and quality of figure make it impossible to read the text in this figure. Please increase the size of font and quality of figure file. I also think you miss an opportunity here. This figure should reflect your original framework to show the reader how to apply the framework. At the moment it hard to determine what the ecosystem state you are trying to describe is and what the drivers are, etc.

Author Response

We would like to express our appreciation to the reviewer 1 for the careful review and constructive suggestions. The manuscript has been revised to address each of the reviewer's comments. For reference to the changes made, please see the marked version of the manuscript.

REV1

General comments

C1

I have three main comments, but the first is the most important to deal with. The paper suffers from a lack of clarity and organization. Examples of this include: 1) over use of bulleted lists that are long and do not highlight important parts; 2) Table 2 is confusing in both format and what points it is trying to get across; and 3) Case study description needs to be streamlined to the important parts of the system for coastal squeeze. These are just a few examples, but I think the authors need to decide on the important points of the paper and try to trim back the rest.

We appreciate these critical comments. We have tried to improve the ms according to all the suggestions, comments and recommendations made by the five referees.

C2

My second comment is that the authors should detail how to use the framework quantitatively. At the moment, it is just description, except for the very short “Parameters” sections. Details on how to apply this system quantitatively is critical for the adoption of this framework by other users. Lastly, the figures really need improvement. They need to be higher resolution and the text needs to be bigger. See specific comments below, but it was hard for me to even understand the figures because I could not read them. Please see below for a list of specific comments on the document.

One of the aspects that we consider important in the ms is a more precise definition of Coastal Squeeze that allows evaluation and decision making for its management. The example is simply illustrative, however, following the recommendations we have included more quantitative information.

All the figures were reworked.

REV1

Specific Comments:

Q1

Stylistic comment – Watch for use of passive voice. I also find there are too many bulleted lists. It is a good way to highlight an important list – but is overused in this paper.

We have reduced the use of bulleted points and a English speaker checked the ms.

Q2

Title – Why is “Coastal Squeeze” capitalized in the title but nowhere else?

Thanks to your comment we decided to write coastal squeeze in lower case letters

Q3

Section 2.1.3 – I find the use of the word extension here confusing. The authors should clarify what they mean by this section.

That section has been incorporated into a smoother paragraph so the subtitle has gone.

Sections 2.1.4, 2.3.3 – I find the parameters section short and somewhat confusing. I think they authors could spend more space explaining how to estimate drivers if this framework is supposed to be quantitative. I think that these parameters sections could be merged into a larger section about quantitatively analyzing these components of the system.

The suggestion has been addressed

Q4

Lines 268-293: This site description section could be shortened and refined to only discuss targeted information. It is too long at the moment.

The suggestion has been addressed

Q5

Table 2 – Table 2 is really confusing and hard to follow. The cells seem to be lists or paragraphs with no clear organization.  Please distill into the most important points and condense to a half page.

The suggestion has been addressed. In fact Table 2 was moved to an Annex

Q6

Figure 1 – the quality of the figure and the colors chosen make it almost impossible to read text. Please convert to higher quality figure file and change the light blue and yellow combination.

The suggestion has been addressed

Q7

Figure 3 – Hard to differentiate the changes in wave climate between panels c and d. Is there a way to highlight it better? Also, in panel a, the legend should all be within the panel rather than spanning across multiple panels at the top.

Figure 3 has been modified according to the recommendation. An additional figure (Figure A, Annex 2) was included in order to quantify the wave height differences between the two scenarios.

Q8

Figure 5 – Size of text and quality of figure make it impossible to read the text in this figure. Please increase the size of font and quality of figure file. I also think you miss an opportunity here. This figure should reflect your original framework to show the reader how to apply the framework. At the moment it hard to determine what the ecosystem state you are trying to describe is and what the drivers are, etc.

Figure 5 has been modified according to the recommendation. 

The aim of this figure is to show the cascading biophysical exchanges. We believe that figure 1 and figure 4 adequately illustrate the framework. Thank you very much for the suggestion

Reviewer 2 Report

This manuscript is well developed and written.

Author Response

We would like to express our appreciation to the reviewer 2 for the constructive evaluation.

Reviewer 3 Report

The paper deals on coastal squeeze indicating a loss of coastal habitats, due to, for instance, sea level rise and the presence of defence structures.

In the introduction, different definitions of coastal squeeze are provided, but the scope of the paper is limited to improve the characterization of coastal squeeze using a modified DPSIR framework: the DES cycle, which is described in section 2 and 3. The section 4 is dedicated to the study case and section 5 to the discussion/conclusion.

The paper does not follow the classical structure and does not include the description of the methodology and a detailed description of the pilot sites. For instance, no information is provided regarding the data used for the case study. Similarly, the scenarios were obtained using the WAPO numerical model, but the authors do not explain the model. As a consequence, it is not clear how the authors have developed the scenarios and what data have been used.

The scenarios represent the results, but the authors should explain how the scenarios have been developed.

In addition, the description of the different scenarios is not supported by data, as for instance l367-370.

A list of eventual measures is provided like restoring seagrasses or executing integral coastal management, but they are generic. The authors do not explain if these measures are well adapted to the pilot sites.

The discussion and conclusion are very poor.

Author Response

We would like to express our appreciation to the reviewer 3 for the careful review and constructive suggestions. The manuscript has been revised to address each of the reviewers' comments. For reference to the changes made, please see the marked version of the manuscript.

Rev 3

General comments

C1

The paper deals on coastal squeeze indicating a loss of coastal habitats, due to, for instance, sea level rise and the presence of defence structures.

Some clarifications have been included as a result of the contributions of the five reviewers.

C2

In the introduction, different definitions of coastal squeeze are provided, but the scope of the paper is limited to improve the characterization of coastal squeeze using a modified DPSIR framework: the DES cycle, which is described in section 2 and 3. The section 4 is dedicated to the study case and section 5 to the discussion/conclusion.

The manuscript now has more information

C3

The paper does not follow the classical structure and does not include the description of the methodology and a detailed description of the pilot sites. For instance, no information is provided regarding the data used for the case study. Similarly, the scenarios were obtained using the WAPO numerical model, but the authors do not explain the model. As a consequence, it is not clear how the authors have developed the scenarios and what data have been used.

In order to address this comment, the structure and content of the ms have been substantially modified

C4

The scenarios represent the results, but the authors should explain how the scenarios have been developed.

The suggestion has been addressed

C5

In addition, the description of the different scenarios is not supported by data, as for instance l367-370.

The suggestion has been addressed. There are now some additional references.

C6

A list of eventual measures is provided like restoring seagrasses or executing integral coastal management, but they are generic. The authors do not explain if these measures are well adapted to the pilot sites

In Puerto Morelos there is no integrated programme for the management of the coast, but we are trying to promote it.

C7

The discussion and conclusion are very poor. 

This section has been improved

Reviewer 4 Report

TITLE
This title is more apt to a review than to a new tool presentation. I would suggest that the authors re-evaluate the title.

ABSTRACT
Lines 17-18 "There are several criteria..." If authors want to present the limits of the existing methods, they should either better emphasise in the next sentence that their method tries to address such issues.

Lines 19-20 "Here we propose [...] The DESCR framework" The way this is written now make unclear that DESCR is the framework the authors are proposing. I encourage the authors to connect the two sentences better.

Lines 20-22 It would be useful to highlight in bold both the acronym "DESCR" and the first letter of the words "Drivers", "Exchanges"., "States", "Consequences", and "Response."

INTRODUCTION
Line 37 "The perspective [...] has been changing" Uninformed readers could fail to understand how things are changed. I suggest to be more explicit, or to remove the reference to change (e.g. changing the narrative to multi-disciplinary)

Line 80 "In this cycle" In my opinion, this paragraph is partly redundant. I would suggest writing here a short introduction (rearranging and integrating lines 97-101 and 94-96), and to distribute the information contained in the bullet points between Figure 1 caption (that is relatively information-poor) and the detailed description of each framework "compartment" below.

Line 250: Is it possible to have a reference (or more details) about this program?

Line 255: Is it possible to add Lat and Lon coordinates for the study site?

Line 324: could the authors add a reference for white syndrome?

Lines 395-465: The author should consider to make this section less schematic, either aggregating bullet points or removing the list structure altogether in favour of long paragraphs.

DISCUSSION AND CONCLUSION

Authors did a good job in synthesising and contextualising their effort. Still, they should provide some more comparison with existing assessments/methods, and be more explicit in describing the advantages of the DESCR framework.

FIGURES

Figure 3: (c) and (d) are challenging to tell apart from each other. One possible way to help the visualisation could be subtracting (c) from (d), and masking the land. I added an image composite made this way:

I encourage the authors to evaluate it and produce their version, or any other thing that could increase reader understanding of the difference between sea-level scenarios.

Figure 4: I want to praise the authors for this hand-drawn panel: I find it aesthetically pleasing and informative. At the same time, I would like authors to explain the meaning of the green colour, and suggest the use of a different shade of yellow for coding "moderate energy" since the present one has low-readability and could be confused with some of the life-forms inhabiting the reef. I am also unable to understand what sailing wedge is and why it shows as an arrow only in Fig. 4d.

Tables: I would move both table 1 and table 2 into Supplementary materials.

Author Response

We would like to express our appreciation to the reviewer 4 for the careful review and constructive suggestions. The manuscript has been revised to address each of the reviewer's comments. For reference to the changes made, please see the marked version of the manuscript.

REV 4

General comments

C1

TITLE
This title is more apt to a review than to a new tool presentation. I would suggest that the authors re-evaluate the title.

The suggestion has been addressed

C2

ABSTRACT
Lines 17-18 "There are several criteria..." If authors want to present the limits of the existing methods, they should either better emphasise in the next sentence that their method tries to address such issues.

The suggestion has been addressed

C3

Lines 19-20 "Here we propose [...] The DESCR framework" The way this is written now make unclear that DESCR is the framework the authors are proposing. I encourage the authors to connect the two sentences better.

The suggestion has been addressed

C4

Lines 20-22 It would be useful to highlight in bold both the acronym "DESCR" and the first letter of the words "Drivers", "Exchanges"., "States", "Consequences", and "Response."

The suggestion has been addressed

C5

INTRODUCTION
Line 37 "The perspective [...] has been changing" Uninformed readers could fail to understand how things are changed. I suggest to be more explicit, or to remove the reference to change (e.g. changing the narrative to multi-disciplinary)

The suggestion has been addressed

C6

Line 80 "In this cycle" In my opinion, this paragraph is partly redundant. I would suggest writing here a short introduction (rearranging and integrating lines 97-101 and 94-96), and to distribute the information contained in the bullet points between Figure 1 caption (that is relatively information-poor) and the detailed description of each framework "compartment" below.

The suggestion has been addressed

C7

Line 250: Is it possible to have a reference (or more details) about this program?

Although the area has been extensively studied, there is no integrated programme to address coastal squeeze on the Mexican coast

C8

Line 255: Is it possible to add Lat and Lon coordinates for the study site?

The suggestion has been addressed

C9

Line 324: could the authors add a reference for white syndrome?

Done

C10

Lines 395-465: The author should consider to make this section less schematic, either aggregating bullet points or removing the list structure altogether in favour of long paragraphs.

The suggestion has been addressed

C11

DISCUSSION AND CONCLUSION
Authors did a good job in synthesising and contextualising their effort. Still, they should provide some more comparison with existing assessments/methods, and be more explicit in describing the advantages of the DESCR framework.

Thank you very much. The suggestion has been addressed

C11

FIGURES

Figure 3: (c) and (d) are challenging to tell apart from each other. One possible way to help the visualisation could be subtracting (c) from (d), and masking the land. I added an image composite made this way:

I encourage the authors to evaluate it and produce their version, or any other thing that could increase reader understanding of the difference between sea-level scenarios.

We have now included a figure in Annex 2 that shows the local changes (residuals) in wave height between the current sea level scenario (Figure 3 c) and that of a 0.5 m sea level rise (figure 3 d)).

C12

Figure 4: I want to praise the authors for this hand-drawn panel: I find it aesthetically pleasing and informative. At the same time, I would like authors to explain the meaning of the green colour, and suggest the use of a different shade of yellow for coding "moderate energy" since the present one has low-readability and could be confused with some of the life-forms inhabiting the reef. I am also unable to understand what sailing wedge is and why it shows as an arrow only in Fig. 4d.

Thank you very much. The recommendations have been taken into account

C13

Tables: I would move both table 1 and table 2 into Supplementary materials.

Thanks. The recommendation has been taken into account

Reviewer 5 Report

The paper is well written and presents a very interesting topic on which more research is undoubtedly necessary. The content is pertinent and necessary and specifying the proposal made with a real case helps to reflect on it. In general, it provides useful information to the scientific community and fits into the lines of the journal.

Overview and general recommendation:

It has two main blocks, one theoretical and one applied. In the case of the theoretical block, important changes are necessary. Some figures and tables need to be improved.

2.1. Major comments:

  1. The discussion on the background should be expanded, specifically on the DPSIR framework that is used as reference.
  2. The proposed alternative framework must be better structured. For now it is somewhat confusing, the elements that compose it are not well differentiated and the examples used to explain them do not help to understand it better.
  3. In general, it is very focused on natural elements and processes and the interrelation with anthropic processes is missing.
  4. It should be better explained if it is useful as a causal framework.
  5. More emphasis should be placed on the advantages it has for decision-making and information organization.
  6. How did you construct your DESCR method?

2.2. Minor comments:

  1. (Page 2, line 53) Table 1: It is an excessively wide table. Maybe you can group those definitions that share perspective and causes and summarize all the definitions in one, pointing out the sources from which it comes.
  2. (Page 3, line 67). The DPSIR model should be further discussed, as a fundamental background for this article. It must be shown that it has been read and reviewed enough to propose an alternative and explain why. Reading this work is recommended, among other possible ones: https://www.frontiersin.org/articles/10.3389/fmars.2016.00177/full
  3. (Page 3, line 67-75) To propose a modification of the well-established DPSIR model, it would have to be discussed a little more, with a little more research on what is wrong with this model. To decide that the new framework proposed is an advantage, a little more should be said about the approach followed in the introduction (ecosystem approach? Integrated vision?). This advantage should be very clear by choosing well the examples used later, and for this, those of anthropic origin must be better differentiated from those of natural origin.
  4. (Page 3, line 74) If CONSEQUENCES means the same as IMPACTS, Why shouldn´t be used this expression that is consolidated and accepted? It is only a doubt
  5. (Page 3, line 76) Figure 1:
  • If you do not put the definition of the other elements of the DESCR, do not include it on CONSEQUENCES either (or include it in the others as well)
  • It would be interesting if the examples in yellow were linked/interrelated between elements of the DESCR. If they had a relationship in each element of the DESCR it could be easier to understand the differences between them, showing the cause-consequences flux. For example, "sea level rise" could has a clear reflection in Exchanges, in State, in Consequences, in Responses, etc.
  • The examples of the figure 2 are added again in the text. Some are different from those posted here, but others are the same. It seems it has no logic to repit things in a figure and in the text. Either the examples are removed from the text and left in the figure or vice versa, or something different is added to the figure.
  • Lastly, “overfighing” or “non-native species” appear in Exchanges in the text (see comment below) but appear in Drivers in Figure 1.
  1. (Page 4, line 79) It is repetitive to see the definition of the elements of DESCR before Figure 1, in Figure 1, in these three points and then in detail in points 2.1, 2.2, etc.
  2. (Page 4, line 87) In the figure, the marked examples appear in the DRIVERS element. I find it difficult to differentiate each element and the examples do not help to understand it better. I suggest again developing a single example for all the elements, interrelating it, to see how they connect with each other. On the other hand, one of the strengths of the DPSIR is that it links human elements with natural ones and here natural drivers predominate.
  3. (Page 6, line 174) I think the definition of Coastal unit is very good, I think the concept of socio-ecological system could be added at some point, taking advantage of the fact that it is already widely accepted for the ecosystem approach.
  4. (Page 6, line 202) I have the feeling that an opportunity gained with ecosystem services is lost by continually talking about consequences on nature and not connecting it with consequences on human well-being. At that point, it stops being something innovative to return to the old PSIR framework
  5. (Page 6, line 206) The Response is not necessarily aimed at modifying the state of the environment. It can be aimed, for example, at modifying the activity and attitude of society (as you yourself point out: education, regulation...).
  6. (Page 6, line 209) Prevention should be added.
  7. (Page 7, line 241-244) Place a bibliographic reference to this assertion
  8. (Page 10, line 117) Table 2: I don't know if this complete table is necessary. Perhaps it could be left complete as an annex and shown here something more visual and quicker to see. Should it be ordered as the DES framework marks (Drivers first)?

Suggestions are detailed in the attached file

Author Response

We would like to express our appreciation to the reviewer 5 for the careful review and constructive suggestions. The manuscript has been revised to address each of the reviewer's comments. For reference to the changes made, please see the marked version of the manuscript.

REV 5

General comment

GC

The paper is well written and presents a very interesting topic on which more research is undoubtedly necessary. The content is pertinent and necessary and specifying the proposal made with a real case helps to reflect on it. In general, it provides useful information to the scientific community and fits into the lines of the journal.

Overview and general recommendation:

It has two main blocks, one theoretical and one applied. In the case of the theoretical block, important changes are necessary. Some figures and tables need to be improved.

Thank you very much. We reworked all the figures.

REV5

Major comments

MC1

The discussion on the background should be expanded, specifically on the DPSIR framework that is used as reference.

The suggestion has been addressed

MC2

The proposed alternative framework must be better structured. For now it is somewhat confusing, the elements that compose it are not well differentiated and the examples used to explain them do not help to understand it better.

The suggestion has been addressed

MC3

In general, it is very focused on natural elements and processes and the interrelation with anthropic processes is missing.  Has it been done?

The suggestion has been addressed

MC4

It should be better explained if it is useful as a causal framework.

The suggestion has been addressed

MC5

More emphasis should be placed on the advantages it has for decision-making and information organization.

The suggestion has been addressed

MC6

How did you construct your DESCR method?

As the interactions between the drivers and the state of the environment are bidirectional, and involve mass and energy flows, the concept of pressures was replaced by that of exchanges in the DPSIR framework. In addition, in decision making we are convinced that it is more effective to analyse the consequences than the impacts. For this we rely on another risk analysis framework that is widely accepted (Source-Pathway-Receptor-Consequence).

REV5

Minor comments

C1

(Page 2, line 53) Table 1: It is an excessively wide table. Maybe you can group those definitions that share perspective and causes and summarize all the definitions in one, pointing out the sources from which it comes.

The suggestion has been addressed

C2

(Page 3, line 67). The DPSIR model should be further discussed, as a fundamental background for this article. It must be shown that it has been read and reviewed enough to propose an alternative and explain why. Reading this work is recommended, among other possible ones: https://www.frontiersin.org/articles/10.3389/fmars.2016.00177/full

The suggestion has been addressed

C3

(Page 3, line 67-75) To propose a modification of the well-established DPSIR model, it would have to be discussed a little more, with a little more research on what is wrong with this model. To decide that the new framework proposed is an advantage, a little more should be said about the approach followed in the introduction (ecosystem approach? Integrated vision?). This advantage should be very clear by choosing well the examples used later, and for this, those of anthropic origin must be better differentiated from those of natural origin.

The suggestion has been addressed

C4

(Page 3, line 74) If CONSEQUENCES means the same as IMPACTS, Why shouldn´t be used this expression that is consolidated and accepted? It is only a doubt

Consequences and impacts are not necessary the same. We have improved the explanation in this section.

C5

(Page 3, line 76) Figure 1:

·         If you do not put the definition of the other elements of the DESCR, do not include it on CONSEQUENCES either (or include it in the others as well)

·         It would be interesting if the examples in yellow were linked/interrelated between elements of the DESCR. If they had a relationship in each element of the DESCR it could be easier to understand the differences between them, showing the cause-consequences flux. For example, "sea level rise" could has a clear reflection in Exchanges, in State, in Consequences, in Responses, etc.

·         The examples of the figure 2 are added again in the text. Some are different from those posted here, but others are the same. It seems it has no logic to repit things in a figure and in the text. Either the examples are removed from the text and left in the figure or vice versa, or something different is added to the figure.

Lastly, “overfighing” or “non-native species” appear in Exchanges in the text (see comment below) but appear in Drivers in Figure 1.

The suggestion has been addressed

C6

(Page 4, line 79) It is repetitive to see the definition of the elements of DESCR before Figure 1, in Figure 1, in these three points and then in detail in points 2.1, 2.2, etc.

The suggestion has been addressed

C7

(Page 4, line 87) In the figure, the marked examples appear in the DRIVERS element. I find it difficult to differentiate each element and the examples do not help to understand it better. I suggest again developing a single example for all the elements, interrelating it, to see how they connect with each other. On the other hand, one of the strengths of the DPSIR is that it links human elements with natural ones and here natural drivers predominate.

The suggestion has been addressed

C8

(Page 6, line 174) I think the definition of Coastal unit is very good, I think the concept of socio-ecological system could be added at some point, taking advantage of the fact that it is already widely accepted for the ecosystem approach.

Thank you very much. The suggestion has been addressed

C9

(Page 6, line 202) I have the feeling that an opportunity gained with ecosystem services is lost by continually talking about consequences on nature and not connecting it with consequences on human well-being. At that point, it stops being something innovative to return to the old PSIR framework

The figure was moved in order to illustrate the exchanges processes.

C10

(Page 6, line 206) The Response is not necessarily aimed at modifying the state of the environment. It can be aimed, for example, at modifying the activity and attitude of society (as you yourself point out: education, regulation...).

The suggestion has been addressed

C11

(Page 6, line 209) Prevention should be added.

The suggestion has been addressed

C12

(Page 7, line 241-244) Place a bibliographic reference to this assertion

The suggestion has been addressed

C13

(Page 10, line 117) Table 2: I don't know if this complete table is necessary. Perhaps it could be left complete as an annex and shown here something more visual and quicker to see. Should it be ordered as the DES framework marks (Drivers first)?

The suggestion has been addressed. Table 1 and Table 2 were moved to Annex 1

Round 2

Reviewer 1 Report

Paper Summary:

The authors worked hard to revise this paper. The writing is much improved from the original manuscript. I am still concerned with the focus of the narrative and how widely applicable this framework is to other systems. I think the fundamental point that needs to be addresses is: How do other managers and researchers measure these drivers and exchanges to apply their system?

Additionally, the authors did not adequately address my concerns about how this framework can be quantitatively assessed. Even the case study does not describe the data needed to address seagrass and coral states or how to measure their feedbacks on each other. In addition, in the introduction the authors point out that the DPSIR framework was criticized for being over simplified and improvement is needed to quantify links. The authors have not done enough to address this criticism in their own framework and case study. The case study should work to quantify at least some of the exchanges shown in figure 5.  

Finally, terms are sometimes used that are not well defined or discussed (such as equilibrium state). In addition, other terms have meanings in multiple fields, such as “state”, that are sometime strictly defined (i.e. ecosystem state). I think adding a bit more about how the authors are defining these terms could help others to use the framework

Specific Comments:

Even though Tables 1 and 2 were moved to the Appendix and revised, the table still have too much information displayed. Table one still has paragraphs in the first column. In table 2, the states still are not consistently discussed. The information in that column needs to be consistently discussed and formatted.

Figure 3 c and d are still hard to differentiate. I think a single map that shows change in wave climate between the two scenarios might be more informative.

Figure 4 is still hard to work through and brings up new questions. What are equilibrium conditions? Are they describing the ecosystem state?

Author Response

Comments to suggestions made by reviewers 1-5

We would like to express our appreciation to all reviewers for their careful examination of our work, and constructive suggestions. In the first round of revisions, the manuscript was modified in an effort to balance the suggestions and recommendations of all reviewers. As a result, four reviewers have already accepted the manuscript in its current form. In this new round of reviews, we have included the suggestions and recommendations made by Reviewer 1, but still respecting the suggestions and recommendations made previously by the other reviewers. For reference to the changes made, please see the marked version of the manuscript.

Response to reviewer 1:

REV1

General comments

C1

The authors worked hard to revise this paper. The writing is much improved from the original manuscript. I am still concerned with the focus of the narrative and how widely applicable this framework is to other systems. I think the fundamental point that needs to be addresses is: How do other managers and researchers measure these drivers and exchanges to apply their system?

Thank you for your comments. We appreciate very much that the reviewer has noticed the improvement in the manuscript.

Although in a narrative way, we use the example of a site in which several ecosystems are present in a very reduced space, we are confident that the framework we are proposing can be applied to other coastal units.

As stated in section 2.1:

·       The way to identify and evaluate drivers is analogous to the way it is done in the DPSIR framework. That is, a driver governs the direction of ecosystem change and could be of human or natural origin. Drivers can be physical, such as non-local currents, tides, waves, winds, tsunamis, runoff, availability of light, temperature, dredging, and beach nourishment. They can also be chemical, such as pollution and acidification, or biological, such as organisms that modify the abiotic environment through their presence or biological activity.

·       In the original DPSIR framework a driver generates a pressure that eventually modifies the state of the environment and the modification of the state does not alter the pressures. However, we consider that there is an exchange of matter and energy between the drivers and ecosystems (among other processes, which is why the pressures generated by drivers on the environment are taken into account here). In other words, when a driver modifies the environment, the exchanges are also affected and eventually the drivers are altered.  Exchanges can be evaluated through locally validated numerical models or parametrizations.

C2

Additionally, the authors did not adequately address my concerns about how this framework can be quantitatively assessed. Even the case study does not describe the data needed to address seagrass and coral states or how to measure their feedbacks on each other. In addition, in the introduction the authors point out that the DPSIR framework was criticized for being over simplified and improvement is needed to quantify links. The authors have not done enough to address this criticism in their own framework and case study. The case study should work to quantify at least some of the exchanges shown in figure 5.

As stated in the introduction: “in this paper, we aim to improve the characterization of coastal squeeze and expand its definition based on multifactorial interacting elements.”

In addition, the introduction includes: “Patrício et al. (2016) reported that since 1999, 25 schemes for management and decision-making across ecosystems have used DPSIR conceptual framework derivations to structure and analyse information. The authors point out that even so, the framework has significant shortcomings. In particular, the assessment of Pressures, State and Impacts oversimplifies environmental problems. The authors cited above recognize that clearer, more comprehensive, nested conceptual models are needed to quantify the links between pressure-state change in marine and coastal ecosystems.” The criticism referred to by the Reviewer emanates from the work of Patrício et al. (2016) and the present manuscript aims to generate a bridge to alleviate such criticism.

Patrício, J.; Elliott, M.; Mazik, K.; Papadopoulou, K.-N.; Smith, C.J. DPSIR—Two Decades of Trying to Develop a Unifying Framework for Marine Environmental Management? Frontiers in Marine Science 2016, 3, 177.

C3

Finally, terms are sometimes used that are not well defined or discussed (such as equilibrium state). In addition, other terms have meanings in multiple fields, such as “state”, that are sometime strictly defined (i.e. ecosystem state). I think adding a bit more about how the authors are defining these terms could help others to use the framework

Terms that have been used by other authors and that we consider to be widely accepted, in previous evaluations carried out and reported in a large number of publications, such as "state", have not been defined again in this manuscript.

In the case of "ecosystem state" we respond in the reply to this reviewer in Specific Comments Q3     

“A schematic summary of the exchanges is presented in figure 5. The equilibrium condition should not be interpreted as a static condition, since any ecosystem needs to grow, mature and regenerate. Dynamic equilibrium is achieved when the exchanges of matter and energy associated with periods of rain and low water, and of periods of calm, as well as exposure to storms (such as hurricanes) give stable conditions.”

REV1

Specific Comments:

Q1

Even though Tables 1 and 2 were moved to the Appendix and revised, the table still have too much information displayed. Table one still has paragraphs in the first column. In table 2, the states still are not consistently discussed. The information in that column needs to be consistently discussed and formatted.

In Table A some of the definitions given in the first column have now been simplified.

In Table B, the explanation referred to has been extended. “The coverage of wetland has been severely diminished by felling, drying, and the modification of water flows, usually to make way for the construction of tourist infrastructure and roads. As a means of illustrating the main drivers, their biophysical exchanges and possible response in the state of the mangrove, dune, beach, seagrasses and coral, additional information on the main characteristics of these ecosystems appears in Table B (Annex1). It is important to note that changes in the state of the environment will eventually generate different exchanges and characteristics of the drivers. Similarly, with different driver intensities and frequencies the exchanges, and therefore the state of the environment, will be different. ”

In relation to the format of Table B, we believe that this format is accepted by MDPI, however we will check this in due course with those responsible for the production of the journal.

Q2

Figure 3 c and d are still hard to differentiate. I think a single map that shows change in wave climate between the two scenarios might be more informative.

In Figure 3e, the local changes in wave height between the present sea level scenario and a sea-level rise of 0.5 m have been included

Q3

Figure 4 is still hard to work through and brings up new questions. What are equilibrium conditions? Are they describing the ecosystem state?

From the question, we assume that Figure 5 is being referred to, and not figure 4.

The following sentence has been included “A schematic summary of the exchanges is presented in figure 5. The equilibrium condition should not be interpreted as a static condition, since any ecosystem needs to grow, mature and regenerate. Dynamic equilibrium is achieved when the exchanges of matter and energy associated with periods of rain and low water, and of periods of calm, as well as exposure to storms (such as hurricanes) give stable conditions.”

Reviewer 3 Report

The paper has been improved in order to be accepted for publication... well done!

Author Response

We would like to express our appreciation for your careful examination of our work, and constructive suggestions.

Reviewer 5 Report

The recommendations have been mostly well attended and additional improvements have been made, some perhaps associated with other reviewers, which improve the initial version of the paper.

A general minor revision should be made, to attend possible minor aspects. I put three as an example:

  1. (Page 2, line 105) After "infrastructures" a point appears that must be removed.
  2. (Page 2, line 107) Correct "Drving" (it should be repleaced by "driving")
  3. (Page 2, line 132) "Made" is repeated in the same sentence
